# Influence of Implant Dimensions and Position on Implant Stability: A Prospective Clinical Study in Maxilla Using Resonance Frequency Analysis

**Antonio Nappo** [1], **Carlo Rengo** [1], **Giuseppe Pantaleo** [2], **Gianrico Spagnuolo** [2,*] and **Marco Ferrari** [1]

1   Department of Prosthodontics and Dental Materials, University of Siena, 53100 Siena, Italy; dr.nappo@libero.it (A.N.); carlorengo@alice.it (C.R.); ferrarm@gmail.com (M.F.)
2   Department of Neurosciences, Reproductive and Odontostomatological Sciences, University of Naples "Federico II", 80138 Naples, Italy; giuseppepantaleo88@gmail.com
*   Correspondence: gspagnuo@unina.it; Tel.: +39-0817462080

**Abstract:** Implant stability is relevant for the correct osseointegration and long-term success of dental implant treatments. The aim of this study has been to evaluate the influence of implant dimensions and position on primary and secondary stability of implants placed in maxilla using resonance frequency analysis. Thirty-one healthy patients who underwent dental implant placement were enrolled for the study. A total of 70 OsseoSpeed TX (Astra Tech Implant System—Dentsply Implants; Mölndal, Sweden) implants were placed. All implants have been placed according to a conventional two-stage surgical procedure according to the manufacturer instructions. Bone quality and implant stability quotient were recorded. Mean implant stability quotient (ISQ) at baseline (ISQ1) was statistically significant lower compared to 3-months post-implant placement (ISQ2) ($p < 0.05$). Initial implant stability was significantly higher with 4 mm diameter implants with respect to 3.5 mm. No differences were observed within maxilla regions. Implant length, diameter and maxillary regions have an influence on primary stability.

**Keywords:** dental implants; osseointegration; resonance frequency analysis

## 1. Introduction

In the literature many authors have proposed advantageous long-term results for implant-supported single-unit crowns, as well as, implant-supported short-span fixed dental prostheses (FDP) [1,2]. Implant success depends on tissue biological response and on several other factors such as smoking habit, periodontal status and surgical technique [3–6]. Primary and secondary stability are determining factors for successful implant osseointegration [7] and the absence of micro-movements is a necessary condition [8–10]. A combination of multiple variables could influence primary implant stability such as:

- The quality and quantity of bone at the recipient site;
- The surgical technique used in order to place the implant;
- The macro-/microscopic morphology of the implant [11–16].

Secondary stability is the progressive increase in stability as a consequence of the dynamic interrelationship between new bone formation and remodelling occurring at the bone-implant interfacial zone [17]. In the literature it is demonstrated that the implant success depends on the quality and quantity of the bone as most important factors [18–20]. Bone resorption and healing delay

are the result of implant failure due to weak bone quantity and quality. Jaffin et al. showed a 35% failure rate in type 4 bone. In their study, the major risk factors for implant failure were weak bone quantity and quality [21]. Moreover, Bischof and co-workers demonstrated that the primary stability of the implant depends only on jaw and the bone type and not on other factors such as diameter, length and deepening of the implant [22].

Therefore, the aim of the present study was to evaluate the influence of implant dimensions (length and diameter) and position (anterior and posterior maxilla) on the primary and secondary stability of implants placed in the upper arch.

## 2. Materials and Methods

### 2.1. Study Design

The study was designed as a prospective clinical trial.

### 2.2. Patient Selection

Patients consecutively treated at the Department of Oral Surgery of the University of Siena and University "Federico II" of Napoli, Italy, were enrolled for the study, the recruitment period of included patients was 12 months. All patients agreed to participate in the study and signed informed consent. Thirty-one patients (mean age: 57; range from 31 to 77) have been enrolled in the study, 17 females (mean age: 56; range from 31 to 73) and 14 males (mean age: 59; range from 31 to 77). The study was conducted according to the principles of the Declaration of Helsinki on experimentation involving human subjects.

The inclusion criteria were as follows:

- Patients were aged 18 years or older;
- Absence of medical history or conditions that could contraindicate surgery;
- 4 to 6 months waiting time were necessary for healing after tooth extraction;
- Presence of sufficient residual alveolar bone volume to achieve primary implant stability without concomitant or previous bone augmentation;
- Full-mouth plaque score (FMPS) < 25% at baseline;
- Full-mouth bleeding score (FMBS) < 25% at baseline;

Exclusion criteria were:

- Tobacco smoking;
- Pregnancy and lactation;
- Bisphosphonates use;
- Untreated periodontal conditions;
- Absence at least of 2 mm of keratinized tissue;
- Lower arches.

### 2.3. Clinical Procedure

Dental implants ("OsseoSpeed TX", Astra Tech Implant System—Dentsply Implants; Mölndal, Sweden) were placed following a two-stage protocol according to the manufacturer's instructions. These kinds of implant have two main features: an exclusive implant surface with a fluoride-treated nanostructure that stimulates early bone formation and provides a firmer bone-implant connection; micro-threads on the neck of the implant that ensure optimal load distribution and optimal stress values. Implants were placed exclusively in the upper jaw. For definition purposes, implants placed in the "anterior" maxilla were meant to replace central and lateral incisors and canines; whereas in the "posterior" maxilla implants were placed to replace premolars and molars. Implants were usually positioned with the implant shoulder at the level of the alveolar bone crest and then covered with the

mucosal flap. All the implants were placed in native bone and without bone regeneration. The torque was measured through the implant motor. The implants, placed with handpiece, had all torque up to 35 ncm. The second-phase surgery was carried out at 3 months. Different implant lengths (9, 11 and 13 mm) and diameters (3.5 and 4 mm) were used. The diameter of the last tool used was based on the diameter of the implants, it was 2.7 for diameter 3.5 and 3.2 for diameter 4. A beta-lactam antibiotic (Amoxicillin) was given to all patients for 5 days post-surgically. The postoperative therapy required good oral hygiene, rinsing with mouth wash containing 0.2% chlorhexidine solution twice a day for four post-operative weeks from the surgery. Sutures were removed at seven days at the surgery. All implants were evaluated with peri-apical x-rays immediately after insertion and after 3 months. Definitive crowns were delivered at 4–6 months post-surgery. All prosthesis were manufactured in order to facilitate oral hygiene procedures. No implant failures were recorded. The bone quality was clinically evaluated using the index of Lekholm and Zarb, in agreement with the radiographic evaluations and the drilling resistance perceived by the clinician operator [23]. Implants distribution according to bone quality is showed in Table 1. Implant stability measurements through resonance frequency analysis (RFA) were performed by a single operator immediately after implant placement in terms of the implant stability quotient (ISQ1) and after 3 months (ISQ2). The ISQ was obtained installing a "Smartpeg" transducer (Integration Diagnostics AB, Göteborg, Sweden) into the fixture and approaching it perpendicularly with the handpiece probe of the Osstell (Integration Diagnostics AB, Göteborg, Sweden) device.

**Table 1.** Implants distribution according to bone quality assessment.

| Bone Quality | I | II | III | IV | Total |
|:---:|:---:|:---:|:---:|:---:|:---:|
| N | 2 | 36 | 29 | 3 | 70 |
| % | 2.9 | 51.4 | 41.4 | 4.3 | 100% |

*2.4. Statistical Analysis*

Descriptive statistics (e.g., means and standard deviation (SD)) were used to present the outcomes. The primary outcome was based on the ISQ. Analysis of variance and Tukey's multiple comparison tests and paired *t*-test were performed. A value of $p > 0.05$ was considered as level of statistical significance.

**3. Results**

Mean ISQ at implant placement (ISQ1) was $75.3 \pm 5.5$ whereas after 3 months (ISQ2) it was statistically significantly higher ($p < 0.05$), with a mean of $79.6 \pm 5.8$ (Figure 1). Descriptive statistics of ISQ values distribution within implant diameter and length, maxilla regions, sex and age are reported in Figures 2–4. Tukey's multiple comparison test of ISQ1 and ISQ2 values for all before mentioned parameters are shown in Tables 2–4.

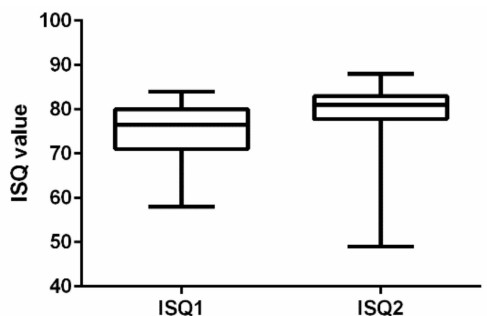

**Figure 1.** Paired t test for implant stability quotient (ISQ) values at implant placement (ISQ1) and after 3 months (ISQ2).

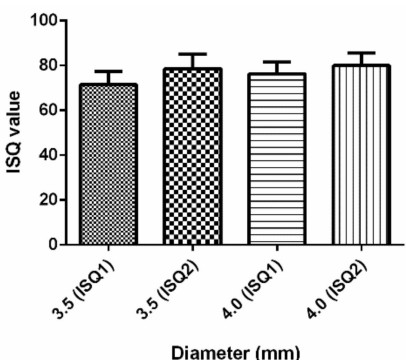

**Figure 2.** ISQ 1 and 2 values distribution within different implant diameters (3.5 and 4 mm).

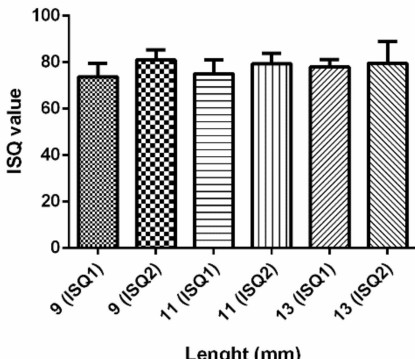

**Figure 3.** ISQ 1 and 2 values distribution within different implant lengths (9, 11 and 13 mm).

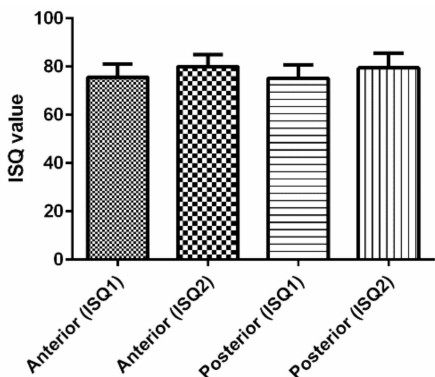

**Figure 4.** ISQ 1 and 2 values distribution within different maxilla regions (frontal and posterior).

**Table 2.** Tukey's multiple comparison test of ISQ1 and ISQ2 values for 3.5- and 4-mm diameter implants. Numbers are means and values in brackets are standard deviations. Lowercase letters indicate statistically significant differences among the diameter within the ISQ 1 or 2 values. Uppercase letters indicate statistically significant differences between the ISQ 1 or 2 values within the diameter.

| Implant Diameter (mm) | ISQ1 | ISQ2 |
| --- | --- | --- |
| 3.5 | 71.38 (5.79) aA | 78.46 (6.43) aB |
| 4 | 76.23 (5.16) bA | 79.89 (5.73) aB |

**Table 3.** Tukey's multiple comparison test of ISQ1 and ISQ2 values for 9-, 11- and 13-mm implants. Numbers are means and values in brackets are standard deviations. Lowercase letters indicate statistically significant differences among the length within the ISQ 1 or 2 values. Uppercase letters indicate statistically significant differences between the ISQ 1 or 2 values within the length. * and ** indicate statistically significant differences between ISQ1 and 2 values across the lengths.

| Implant Length (mm) | ISQ1 | ISQ2 |
|---|---|---|
| 9 | 73.5 (5.91) aA * | 80.93 (4.28) aB ** |
| 11 | 74.95 (5.95) aA ** | 79.25 (4.49) aB * |
| 13 | 77.88 (3.13) abA | 79.44 (9.31) aA |

**Table 4.** Tukey's multiple comparison test of ISQ1 and ISQ2 values for different maxilla regions (frontal and posterior). Numbers are means and values in brackets are standard deviations. Uppercase letters indicate statistically significant differences between the ISQ 1 or 2 values within the maxillary region. * indicates statistically significant differences between ISQ 1 and 2 values across the maxillary regions.

| Maxilla Region | ISQ1 | ISQ2 |
|---|---|---|
| Frontal | 75.59 (5.57) | 80.05 (4.97) * |
| Posterior | 75.21 (5.62) A * | 79.44 (6.24) B |

*3.1. Differences in Implant Stability Quotient (ISQ) Values According to Implant Diameter*

At implant placement ISQ values were statistically significantly higher for 4 mm diameter implants (76.23 ± 5.16) compared with 3.5 mm diameter implants (71.38 ± 5.79) (Table 2).

*3.2. Differences in ISQ Values According to Implant Length*

At implant placement, ISQ values progressively increased with implant length, even if no statistically significant differences were observed within the groups (9, 11 and 13 mm). After 3 months, ISQ values were statistically significantly higher than baseline (ISQ1), but exclusively for 9- and 11-mm length implants. Implants with 13 mm length showed no differences between ISQ1 and ISQ2 values (Table 3).

*3.3. Differences in ISQ Values According to Maxilla Regions*

Both ISQ values at baseline and after 3 months post-implantation were found to be comparable for implants placed in anterior or posterior maxilla, although implants placed in posterior maxilla showed a significant increase of ISQ values after 3 months compared to baseline. No statistically significantly differences were observed within frontal and posterior maxilla at implant placement and after 3 months. ISQ values after 3 months compared to implant placement were statistically significantly higher exclusively for implants placed in the posterior maxilla (Table 4).

## 4. Discussion

This study has evaluated the influence of implant dimensions (length and diameter) on the primary and secondary stability of implants placed in the upper arch.

The results of the present study suggest that the ISQ values significantly increase during the three months of follow-up. The findings also include significant differences between some parameters analysed (implant diameter, implant length and maxilla regions).

Bone density seems to strongly influence implant stability and long-term success, as demonstrated by the higher implant survival rates in the mandible compared to the maxilla, especially the posterior maxilla [24]. Bone density can be objectively measured with different methods, including microCT [25] that may define the bone quality, even if the concept of "bone quality" is not clearly defined in literature.

Currently the most accepted method to assess the bone quality is the one proposed by Lekholm and Zarb [23], which give a scale from 1 to 4 based on the amount of cortical and trabecular bone

evaluated in preoperative radiographs and the tactile sensation of resistance experienced by the clinician during the drilling procedure of implant site preparation. This method, however, is rather subjective [23]. Depending on the bone quality, surgeons may adapt the surgical protocol in order to increase implant primary stability. Adapted surgical protocols includes the preparation of undersized implant size, the use of osteotomes for bone condensation, the use of different specific drills such as countersink or screw tap drills [26–28].

Implant stability can be measured clinically with different more objective quantitative methods, such as the insertion torque, or electronic devices such as the Periotest (Medizintechnik Gulden, Germany) and the Osstell (Integration Diagnostics, Sweden). The insertion torque provides a reliable assessment of the implant stability, but it can be evaluated only at the implant insertion time and cannot be repeated in the follow-up. A recent systematic review concluded that there is no correlation between ISQ and insertion torque values [29]. The Periotest device produces vibrations on the implant and gives a value (PTV) from 8 to 50, while the Osstell device profits from RFA and gives a value (ISQ) from 1 to 100.

Among these two devices the first one raised some criticism since it seems to have a lower sensitivity and is more susceptible to the operator [30].

Implant stability quotient values recorded at the time of implant installation do not reflect the nature of the bone/implant interface and hence the degree of mechanical anchorage. Primary stability may not only be influenced by bone volumetric density and/or bone trabecular connectivity but also by the thickness and density of the cortical layer of the alveolar bone crest. Concerning the bone quality, the outcomes of the present study are in agreement with those presented by Degidi and co-workers, who reported that bone quality does not appear to be crucial for gaining high ISQ values [31], and the low association between bone quality and ISQ values has been demonstrated in clinical studies [32,33]. Moreover, Barewal [34] observed the correlation between the RFA and the Lekholm and Zarb classification and showed a difference only between bone types 1 and 4. By contrast, Östman [35] demonstrated a close correlation between bone quality and RFA values. Huang [36] demonstrated a direct correspondence between bone quality around implants and calculated a decreasing frequency trend. Moreover, Friberg [27] revealed a correlation between bone quality and implant stability measuring the cutting torque and RFA values during implant placement. Furthermore, it has been reported that this is due to the fact that cortical bone is 10 to 20 times stiffer than trabecular bone [37].

## 5. Conclusions

In conclusion, in this study no statistically significant differences in ISQ values were found in terms of different maxillary areas and, therefore, between different bone quality.

Within the limitations of this study, we conclude that some parameters such as implant length and diameter may influence only the primary stability. However, additional controlled and comparative studies are needed to confirm or refute these findings.

**Author Contributions:** All the authors contributed to this study. A.N., C.R., M.F.: conceptualization, funding acquisition, project administration, supervision, writing of review and editing; C.R., G.P.: investigation, writing of the original draft; A.N., G.S., M.F.: investigation.

**Funding:** This research received no external funding.

**Conflicts of Interest:** The authors declare no conflict of interest.

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
