# Peer review of "Influence of Implant Dimensions and Position on Implant Stability: A Prospective Clinical Study in Maxilla Using Resonance Frequency Analysis"

_applsci, doi:10.3390/app9050860_

Round 1

Reviewer 1 Report

Dear Authors,

the study has been well designed, and it is scientifically sound.
Only a few minor changes are required before it can be accepted for publication:

-        The manuscript needs a slight English language revision from a native speaker, for some grammar and punctuation details.

-        Page 2, line 54: “perspective clinical trial”, “prospective”.

-        Page 2, line 56: As per well-described prospective studies, Period of recruitment (besides the place of recruitments) of included patients should be reported.

-        Given the substantial number of perimenopausal/menopausal female subjects included, could you please clarify if osteoporosis/osteopenia has been taken into account during your patient recruitment?

-        explain why these types of implants have been chosen and what are the advantages or disadvantages in terms of design and morphology of the coils.

-        Lastly, specify if all implants were inserted with the same torque even if the bone quality was different. If so, specify the degree of torque used and which guidelines have been followed.

Author Response

Please find our response to the reviewer's comments in the file 

Thank you

Reviewer 2 Report

Dear Authors 

Few editions of studies required an introduction and discussion heading;

a) https://www.sciencedirect.com/science/article/pii/B9780081021965000112 

b) Keywords are very limited try to add those which enhance the visibility of the paper to the researchers and clinicians.

c)  Check line 154-156, some typing mistake. 

d) Authors can expand conclusion heading, discuss a few more lines from the experiment outcomes. 

e) Check References number: 13, 17, 37

Author Response

(The authors gave the same response as above.)

Reviewer 3 Report

Although the manuscript is well written, my only comment would be for the novelty of the subject. There are many other studies showed the same outcomes.

Author Response

(The authors gave the same response as above.)

Reviewer 4 Report

The concept of work is based on the state of knowledge from over 10 years ago.

Citations 17 - 22 : 1991- 2006 .

Did not the Authors find anything in this topic in the last decade and need to provide a list of works from recent years?

With this introduction, it is difficult to prove that the work is innovative.

In addition, the results cited in the papers 17-22 also concern partial dentures, which should be definitely divided for many reasons from single tooth loss.

The main reason is the denture biomechanics that affects stability.

In clinical procedure the hole preparation was not described. What was diameter of last tool ? Was the template used or not ?

Line 86: All 86 implants were evaluated with peri-apical x-rays

This is not precise. What was rated.  What about marginal bone crest level ?

Why was not the standardized measurement of marginal bone level and loss applied?

The vertical dimension crown+abutment above initial bone level was not defined.

Line 88: Definitive crowns were delivered

The vertical length of (crown+abutment) above initial bone crest level was not defined.

Line 91: evaluations and the drilling resistance felt perceived by the clinician operator.

Has any scale been adopted? Have these sensations been tested on materials of known density / hardness

Line 92

The description of the measurement technique and the scheme can be helpful to readers who do not know this method and the ISQ scale.

Line 105 and 106 in "Results" belongs to "Methods"

Line 154. The results of the present study suggest that the ISQ values significantly increase during the  three months of follow-up.

Yes, medium. However, it can be seen in Fig. 1 that ISQ for many of implants decreases from almost 60 to just 50. Initial stability was clearly less dispersive.  Many implants have lost their initial stability. Why this has not been discussed ?

My conclusion:

the most interesting thing that the lower band of ISQ was decreased is not shown.

The most interesting was for which implants this fall occurred and whether it depended on anything? Because the fact that statically is a certain average increase is widely known, because as the authors cite, this has been demonstrated in various works.
So, something new can be demonstrated by the Authors by looking at the results and showing the reasons for the decrease of the ISQ lower band (level) after healing period.

Author Response

(The authors gave the same response as above.)

Round 2

Reviewer 2 Report

Dear Author 

Check any English mistakes and typing errors. 

Reviewer 4 Report

The Authors did not answer in the discussion an interesting question, why the mobility for some implants increased.

However, I do not want to be an obstacle in the publication, hence I find work is suitable.

It is difficult for me to oppose the standard, which is that since it is good on average, let's not think about cases where something is worse.

However, the work contains some errors that should be corrected.

I think that my work with the manuscript is finished.

Line 97 Definitive crowns were delivered at 46 months  post-surgery. All prosthesis were manufactured in order to facilitate oral hygiene procedures. No 98 implant failures were recorded.

1. If the work does not concern the insertion of the prosthesis, then the methodology is misleading and the above text should be removed as it is not related to the performed study.

Line 90. had all torque up to 35 ncm

Ncm.